# Obstetric admission and maternal mortality in the intensive care unit in Africa: A systematic review and meta-analysis

Alemwork Abie[1]*, Molla Getie Mehari[2], Tenagnework Eseyneh Dagnew[3], Amare Mebrat Delie[3], Mihret Melese[4], Liknaw Workie Limenh[5], Nigus Kassie Worku[6], Eneyew Talie Fenta[3], Dereje Esubalew[7], Mickiale Hailu[8]

**1** Department of Midwifery, College of Medicine and Health Science, Bahir Dar University, Bahir Dar, Ethiopia, **2** Department of Medical Laboratory Science, College of Medicine and Health Sciences, Injibara University, Injibara, Ethiopia, **3** Department of Public Health, College of Medicine and Health Sciences, Injibara University, Injibara, Ethiopia, **4** Department of Human Physiology, School of Medicine, College of Medicine and Health Science, University of Gondar, Gondar, Ethiopia, **5** Department of Pharmaceutics, School of Pharmacy, College of Medicine and Health Sciences, University of Gondar, Gondar, Ethiopia, **6** Department of Public Health, College of Medicine and Health Science, Dire Dawa University, Dire Dawa, Ethiopia, **7** Department of Human Physiology, College of Medicine and Health Science, Ambo University, Ambo, Ethiopia, **8** Department of Midwifery, College of Medicine and Health Science, Dire Dawa University, Dire Dawa, Ethiopia

* abiealemwork84@gmail.com

## Abstract

### Background

Obstetric complications are a major contributor to maternal morbidity and mortality world-wide, especially in low-resource settings such as many countries in Africa. Intensive care units offer specialized care for women with severe obstetric complications, which require advanced monitoring and therapeutic interventions. Despite the critical importance of ICU care, there is a paucity of comprehensive data on obstetric ICU admissions and outcomes in Africa. Therefore, this study aimed to assess the proportion of obstetric admissions and maternal mortality in the intensive care unit in Africa.

### Objective

To assess the proportion of obstetric admissions and maternal mortality in the intensive care unit in Africa.

### Methodology

The Preferred Reporting Items for Systematic Reviews and Meta-Analysis (PRISMA 2020) guidelines were followed in reporting the study's findings. Articles searched; from PubMed, HINARI, Web of Science, Google Scholar, Repository, and African Journals Online were systematically searched for reports of obstetric admission in the intensive care unit, either pregnant or up to 6 weeks postpartum. The Joanna Briggs Institute (JBI) Critical Appraisal tool was used to evaluate each article's quality. The Stata 17 software was used to compute the analysis. The heterogeneity of the studies was detected using the Cochran Q

**Data availability statement:** All relevant data are within the paper and its Supporting Information files.

**Funding:** The author(s) received no specific funding for this work.

**Competing interests:** All authors declare that they have no competing interests.

**Abbreviations:** ICU, intensive care unit; CI, confidence interval; HDU, high-dependency unit; HDP, hypertensive disorders of pregnancy; JBI, Joanna Briggs Institute.

test and I² test statistics, which were considered significant at p < 0.05. The random effect model of analysis was used with evidence of heterogeneity. Egger's test at p < 0.05 was used to check the evidence of publication bias within the studies. Subgroup analysis and sensitivity analysis were done.

## Result

Eleven studies were included in this study with a total of 10,190 mothers admitted to the intensive care unit. The overall pooled proportion of obstetric intensive care unit admissions in Africa was 17.22% (95% CI; 12.97–21.47; $I^2$ = 97.63%). This translates to roughly 17 out of every 100 ICU admissions being for obstetric complications. Hypertensive disorders of pregnancy 42.96% (95% CI: 27.3, 58.56) and obstetric hemorrhage 24.15% (95% CI: 18.12, 30.18) were the common indications for obstetric admission in the intensive care unit.. Maternal mortality among ICU-admitted patients reached a concerning 30.69% (95% CI: 23.16, 38.22; $I^2$= 93.34%). This means that, nearly one in three women admitted to the ICU for obstetric complications died.

## Conclusion

In Africa, the proportion of obstetric admissions and maternal mortality in the intensive care unit is significant. This high percentage of obstetric admissions and maternal mortality in the ICU highlights the necessity to enhance emergency obstetric care services and invest in the development of well-equipped obstetric ICUs to reduce maternal mortality.

## Registration

CRD42024516612.

## 1. Introduction

Maternal Mortality continues to be a critical challenge for women of reproductive age across the African Region. While there has been a global reduction in the maternal mortality ratio (MMR) to 34.2% between 2000 and 2020, the situation remains a disaster in the African region. Notably, over two-thirds (69%) of maternal deaths occur in this region, with 531 deaths per 100,000 live births. The lowest recorded rate is 3 per 100,000 live births in Seychelles, while the highest is observed in Nigeria, at 1,047 per 100,000 live births [1]. Obstetric complications are a major contributor to maternal morbidity and mortality worldwide, especially in low-resource settings such as many countries in Africa [2].

The management of severe obstetric complications often necessitates admission to intensive care units (ICUs) to provide critical care [2]. Intensive care units (ICUs) have emerged as critical components of modern healthcare systems, providing life-saving interventions for critically ill patients. In the context of obstetrics, ICUs offer specialized care for women with severe complications such as postpartum hemorrhage, sepsis, eclampsia, and other life-threatening conditions, which require advanced monitoring and therapeutic interventions [3]. Importantly, the admission of high-risk obstetric patients to ICUs is associated with improved maternal outcomes, suggesting that effective monitoring and timely intervention can reduce morbidity and mortality [4].

In recent decades, there has been a significant increase in the utilization of ICUs globally, particularly in developing countries. However, the rate of obstetric admissions to ICUs varies

significantly due to socioeconomic conditions, healthcare infrastructure, the availability of specialized services, criteria for ICU admission, and the availability of a high-dependency unit (HDU). In developing nations, 7% of intensive care unit admissions are accounted for by critically ill obstetric patients; in developed nations [5], it ranges from 0.08 to 0.76% of deliveries [6,7]. Within Africa, this variation is evident, with obstetric ICU admissions nearly 7% in South Africa [8], 23% in Malawi [9], 26% in Ghana [10], and 29% in Nigeria [11]. The most common indications for ICU admission among obstetric patients include hypertensive disorders of pregnancy (HDP) and obstetric hemorrhage, alongside other serious conditions such as sepsis and cardiac disease [12,13].

There is also a striking connection between the number of maternal deaths and the accessibility of ICU care since the countries with the highest number of maternal deaths are also those with the lowest number of beds per capita in ICUs [14]. High-income countries report 0% to 4.9% of admissions, while low- and middle-income countries report 2% to 43.6% [15].

Maternal mortality remains a devastating public health crisis in Africa, and its reduction is a central goal of the Sustainable Development Goals (SDGs) and national health strategies across the continent. While progress has been made, many countries are still far from achieving the SDG target for maternal mortality [16]. Improving access to high-quality emergency obstetric care, including critical care services, is essential to accelerate progress [17]. However, a significant gap exists in understanding the burden of critical illness among obstetric patients in the continent. The available studies are often limited in scope, focusing on single-center reports or specific countries, resulting in fragmented knowledge. Furthermore, the heterogeneity of healthcare systems, resources, and patient populations across Africa makes it difficult to generalize findings from individual studies. The lack of comprehensive data hinders the development of evidence-based policies and programs to strengthen maternal health services and reduce maternal mortality. Conducting a systematic review and meta-analysis is justified by the need to consolidate data from diverse studies to achieve a comprehensive understanding of the epidemiology of severe obstetric morbidities. This consolidated information is crucial for identifying gaps in the current healthcare system, developing targeted interventions (e.g., improved ICU capacity, enhanced training programs), informing resource allocation decisions, guiding future research priorities, and supporting evidence-based policy decisions aimed at improving maternal health services and outcomes. Additionally, the study will contribute to the global understanding of maternal health challenges in low-resource settings and support efforts to achieve sustainable development goals related to maternal health.

## 2. Method and materials

### 2.1. Search strategy

The protocol for this study has been registered on the International Prospective Register of Systematic Reviews (PROSPERO), the University of York Center for Reviews and Dissemination (https://www.crd.york.ac.uk/prospero/display_record.php?RecordID=516612) with registration number CRD42024516612. The Preferred Reporting Items for Systematic Reviews and Meta-Analysis (PRISMA 2020) [18] guidelines were followed in reporting the study's findings. The databases (PubMed, HINARI, Web of Science, Google Scholar, Repository, and African Journals Online) were searched to identify studies conducted from January 2009 to December 2023. To identify studies pertinent to our study population, the search was conducted using the keywords "obstetric admission," "intensive care unit," and "Africa." For each keyword, relevant Medical Subject Headings (MeSH) terms and entry words were utilized. All the MeSH terms and entry words were combined using "OR" to retrieve a broad range of studies. Then, the four keywords were combined with each of the elements using "AND" to generate specific

and relevant articles (the detail is found in S3 File). All authors independently reviewed all citations retrieved from the electronic search to identify potentially relevant studies.

## 2.2. Eligibility criteria

### 2.2.1. Inclusion criteria.

(1) Study Design: Observational studies reported the proportion of obstetric admissions in the ICU were included.

(2) Study Population: Participants could be pregnant or up to six weeks postpartum.

(3) Publication: Research articles that were both published and unpublished (preprint) were utilized.

(4) Language: Articles written only in the English language were included.

(5) Year: Articles conducted from January 2009 to September 2023 were included.

(6) Area: Articles conducted in African countries

### 2.2.2. Exclusion criteria.
Letters, reviews, editorial reports, case studies, articles lacking an abstract and full text, and duplicate studies were excluded. Additionally, studies that used a different definition, such as the proportion of obstetric patients among total deliveries, were excluded to ensure consistency in the measurement of the primary outcome.

## 2.3. Data extraction and quality assessment

The data was extracted from various databases using a standard Microsoft Excel spreadsheet. Two authors (AA and MGM) independently extracted and reviewed all the articles included in this study. Any disagreement was handled by another two authors (TED and AMD). Finally, consensus was reached through discussion between the authors. The Joanna Briggs Institute (JBI) Critical Appraisal tool [19]; a standardized critical evaluation method, was used to evaluate each article's quality. Each included article's quality was then categorized as high (80% or higher), moderate (65%–80%), or low (less than 65%). Articles rated as low quality were excluded from the analysis to ensure that the findings are based on studies that meet acceptable methodological standards. All authors independently assessed the articles for inclusion in the review. Any disagreement was resolved through discussion after the same procedure was repeated.

## 2.4. Data processing and analysis

The data was entered using a standard Microsoft Excel spreadsheet. Stata 17 software was used to conduct the meta-analysis. The pooled proportion of each study was reported with a 95% confidence interval (CI) using forest plots. Cochran's Q test assesses whether the observed variance is greater than what would be expected by chance. The $I^2$ statistic quantifies the percentage of total variation across studies that is due to heterogeneity rather than chance. A p-value < 0.05 for Cochran's Q test, along with an $I^2$ value greater than 50%, was considered indicative of substantial heterogeneity. In the presence of substantial heterogeneity, a random-effects model with DerSimonian-Laird weights was employed to pool the proportions. The random-effects model accounts for both within-study and between-study variability, providing a more conservative estimate when heterogeneity is present.

To look for publication bias, the funnel plot and the Egger regression test were employed. Egger's test examines the asymmetry of the funnel plot, with a p-value < 0.05 considered indicative of statistically significant publication bias. Subgroup analyses were conducted to explore

potential sources of heterogeneity and examine variations in obstetric ICU admissions across different categories. The subgroup analysis was carried out based on (1) year of publication (before 2020 Vs. 2020 and later), (2) study design (retrospective Vs. prospective), and (3) region where the study was conducted (East Africa, West Africa, and South Africa) because these factors could influence trends over time, reporting biases, and geographical differences in disease burden, respectively. Additionally, a leave-one-out sensitivity analysis was performed to assess the robustness of the pooled estimate and to identify any single study that might be unduly influencing the overall results. This involved iteratively removing one study at a time and recalculating the pooled estimate. Mean substitution was used to impute missing data within individual studies. However, we recognize that mean substitution can underestimate variability and potentially distort relationships between variables.

### 2.5. Ethics approval and consent to participate

This systematic review and meta-analysis adhered to established ethical principles for such reviews. No primary data were collected; instead, publicly available, aggregated data were extracted from published, peer-reviewed articles and one preprint. Consequently, no direct interaction with human participants occurred, eliminating the need for informed consent. Notably, the included preprint reported ethical approval from the IRB of Saint Paul's Hospital Millennium Medical College, Addis Ababa, Ethiopia, on February 24, 2020, with a waiver of informed consent granted for the retrospective review of medical records. This review used aggregated, anonymized data, precluding any attempt to identify individuals and focusing on synthesizing existing evidence regarding.

## 3. Result

### 3.1. Characteristics of the eligible studies

Both published and unpublished researches on obstetric ICU admission in Africa were included in this review. A total of 2375 articles were found during the review. From the total papers, 159 duplicates and 1123 articles not related to the research question were identified and excluded. During the screening of the titles, 618 articles were excluded, and the remaining 478 (475 from database searching and 3 from other sources) articles were screened by abstract, of which 45 articles were assessed for eligibility. Finally, 11 articles were considered appropriate and eligible for analysis to undergo the final systematic review and meta-analysis (Fig 1).

### 3.2. Characteristics of the included studies

Ten studies met the inclusion criteria to undergo the final systematic review and meta-analysis. Studies were conducted in different African countries: five studies were from Nigeria [11–14], one study was from Ghana [15], two studies were from South Africa [16,17], one study was from Ethiopia [18], one study was from Malawi [19] and one study was from Rwanda [20] with a total study population of 10,190 ICU admissions. Regarding the study design, eight of them were retrospective, and two of them were prospective studies. Based on the JBI critical assessment checklist, the included studies' quality varied from moderate to high quality (Table 1).

### 3.3. Proportion of obstetric admission to the ICU in Africa

According to this study, the overall pooled proportion of obstetric ICU admissions in Africa was 17.22% (95% CI: 12.97, 21.47; $I^2$ =97.63%). This finding highlights the significant burden of obstetric emergencies on healthcare systems in the region. This means that, across the studies included in this review, roughly 17 out of every 100 women admitted to the ICU were there

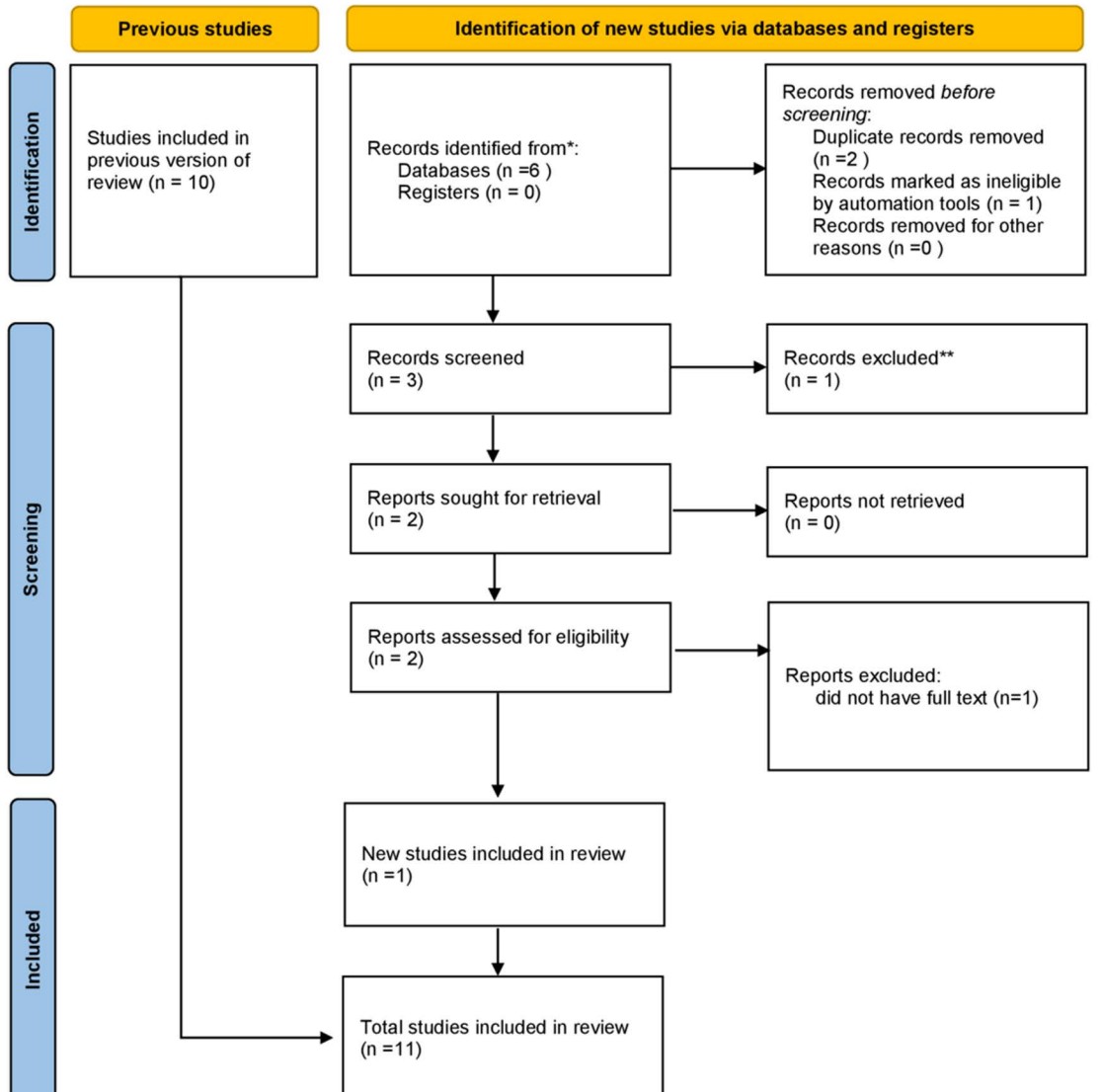

**Fig 1. PRISMA of study selection for systematic review and meta-analysis of obstetric ICU admission.**

due to obstetric complications. However, the high level of heterogeneity ($I^2$ =97.63%) indicates substantial variability in the proportion of obstetric ICU admissions across different settings. It is presented with a forest plot (Fig 2).

## 3.4. Maternal mortality among obstetrics patients in the ICU in Africa

This systematic review and meta-analysis found that, the overall pooled maternal mortality among obstetric ICU admissions in Africa was 30.69% (95% CI: 23.16, 38.22). This indicates a significant burden of maternal mortality associated with critical obstetric illnesses. This translates to nearly one in three women admitted to the ICU for obstetric complications dying. The high level of heterogeneity ($I^2$ = 93.34%) among the included studies suggests substantial

**Table 1. Study characteristics included in the systematic review and meta-analysis in Africa (n=10).**

| Authors | Publica-tion year | data collec-tion period | Country | Study design | Sample size | ICU obstet-ric patients | Proportion of ICU obstetric admission | Quality |
|---|---|---|---|---|---|---|---|---|
| Author A, et al. [20] | 2021 | 10 years | Ghana | Retrospective descriptive | 1721 | 443 | 25.7% | High |
| Author B, et al. [21] | 2022 | 5 years | Nigeria | Retrospective descriptive | 671 | 170 | 25.3% | Moderate |
| Author C, et al. [22] | 2016 | 14 years | Nigeria | Retrospective descriptive | 1336 | 231 | 17.2% | Moderate |
| Author D, et al. [11] | 2015 | 5 years | Nigeria | Retrospective descriptive | 349 | 101 | 28.9% | Moderate |
| Author E, et al. [23] | 2022 | 5 years | Nigeria | Retrospective descriptive | 917 | 118 | 15.4% | Moderate |
| Author F., et al. [8] | 2015 | 4 years | S/Africa | Retrospective study | 2073 | 138 | 6.70% | Moderate |
| Author G., et al. [24] | Preprint | 5 years | Ethiopia | Retrospective cross-sectional | 1200 | 154 | 13% | Moderate |
| Author H. [25] | 2017 | 4 years | S/Africa | Retrospective study | 720 | 210 | 11.6% | High |
| Author I., et al. [9] | 2019 | 1.5 years | Malawi | Prospective observational study | 456 | 105 | 23% | Moderate |
| Author J., et al. [26] | 2021 | 1 year | Rwanda | Prospective cross-sectional | 747 | 94 | 12.6% | Moderate |
| Author K, et al. [27] | 2016 | 4 years | Nigeria | Prospective observational | 870 | 101 | 11.6% | Moderate |

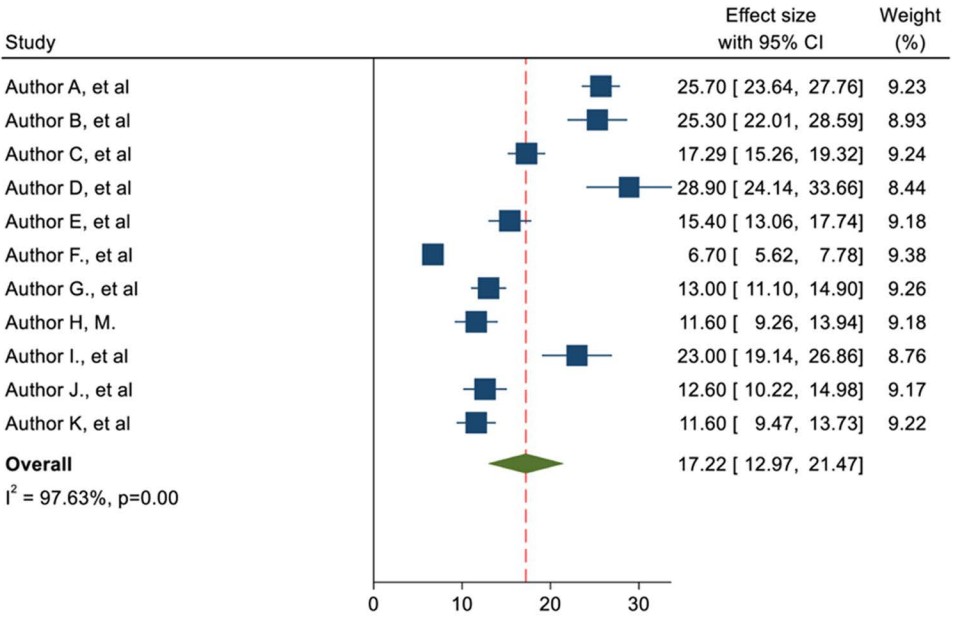

**Fig 2. Forest plot of the pooled proportion of obstetric admission in ICUs in Africa.**

variability in maternal mortality rates across different settings and populations. This finding is illustrated in detail through a forest plot (Fig 3).

### 3.5. Subgroup analysis of obstetric admission and maternal mortality in ICU

Given the marked evidence of heterogeneity observed in proportion of obstetric admission ($I^2$ =97.63%) and maternal mortality ($I^2$= 93.34%) and, a subgroup analysis was conducted considering possible sources of variation, including publication year, region, and study design.

A subgroup analysis was conducted based on study design, categorizing studies as prospective and retrospective studies. A higher proportion of ICU obstetric admission was

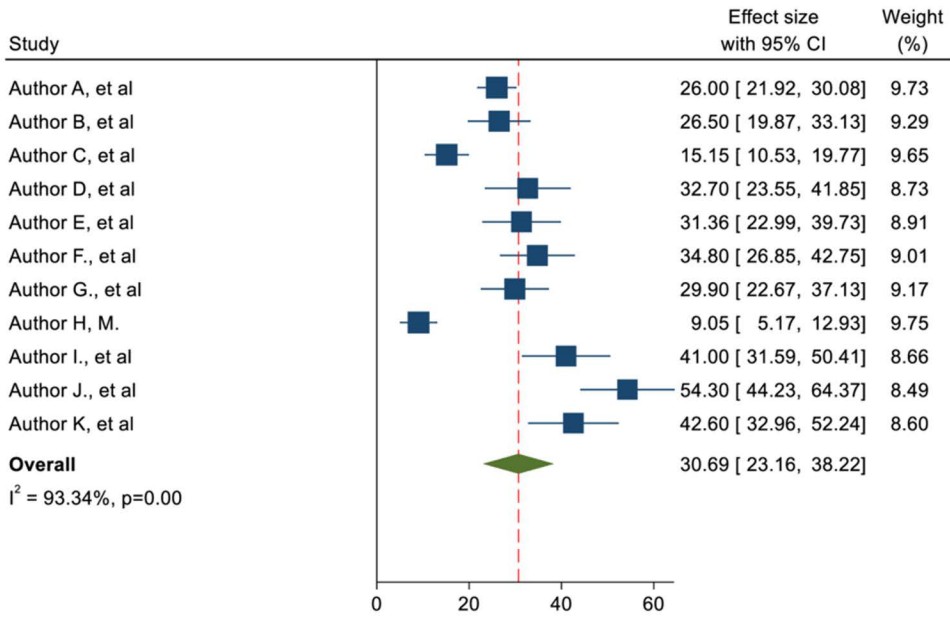

**Fig 3. Forest plot of the maternal mortality among obstetric patients in the ICU in Africa.**

observed in studies with a retrospective study design, reported as 17.85 (95% CI: 12.33, 23.37; $I^2$=98.23%) but the higher proportion of maternal mortality was observed in prospective studies, 45.81% (95% CI: 37.73, 53.89; $I^2$=51.89%) (Fig 4A and B).

Another subgroup analysis was conducted based on publication year, categorizing studies as published before 2020 and after 2020, which showed a higher proportion of ICU obstetric admission and maternal mortality among studies conducted after 2020, 18.36% (95% CI: 12.62, 24.09), and 32.84% (95% CI: 24.88, 40.80; $I^2$=85.24%) respectively (Fig 4C and D). Additionally, subgroup analysis was done based on region (categorizing it as East Africa, West Africa, and South Africa) since there is no adequate study in each country to do subgroup analysis based on country, which revealed the highest proportion of ICU obstetric admission in W/Africa, which was 20.53% (95% CI: 15.35, 25.71%; $I^2$=96.14%) but the highest proportion of maternal mortality in East Africa, which was 41.84% (95% CI: 17.93, 65.74; $I^2$=93.28%) (Fig 4E and F)

### 3.6. Publication bias

A funnel plot was used to visually assess the presence of publication bias; it showed an asymmetric distribution of obstetric admissions in the ICU, suggesting the potential presence of publication bias. Additionally, Egger's test was used to formally assess it, and the results were statistically significant (p = 0.0003), further supporting the presence of this bias (Fig 5).

### 3.7. Sensitivity analysis

To explore the possible cause of the heterogeneity seen in the pooled proportion of obstetric admissions in the ICU, a leave-one-out sensitivity analysis was carried out, indicating that our results were not reliant on a single study. Following the deletion of a single study, the pooled proportion ranged from 16.14% (95% CI: 11.87, 20.41; p = 0.000) to 18.25% (95% CI: 14.57, 21.94; p = 0.000) (Table 2).

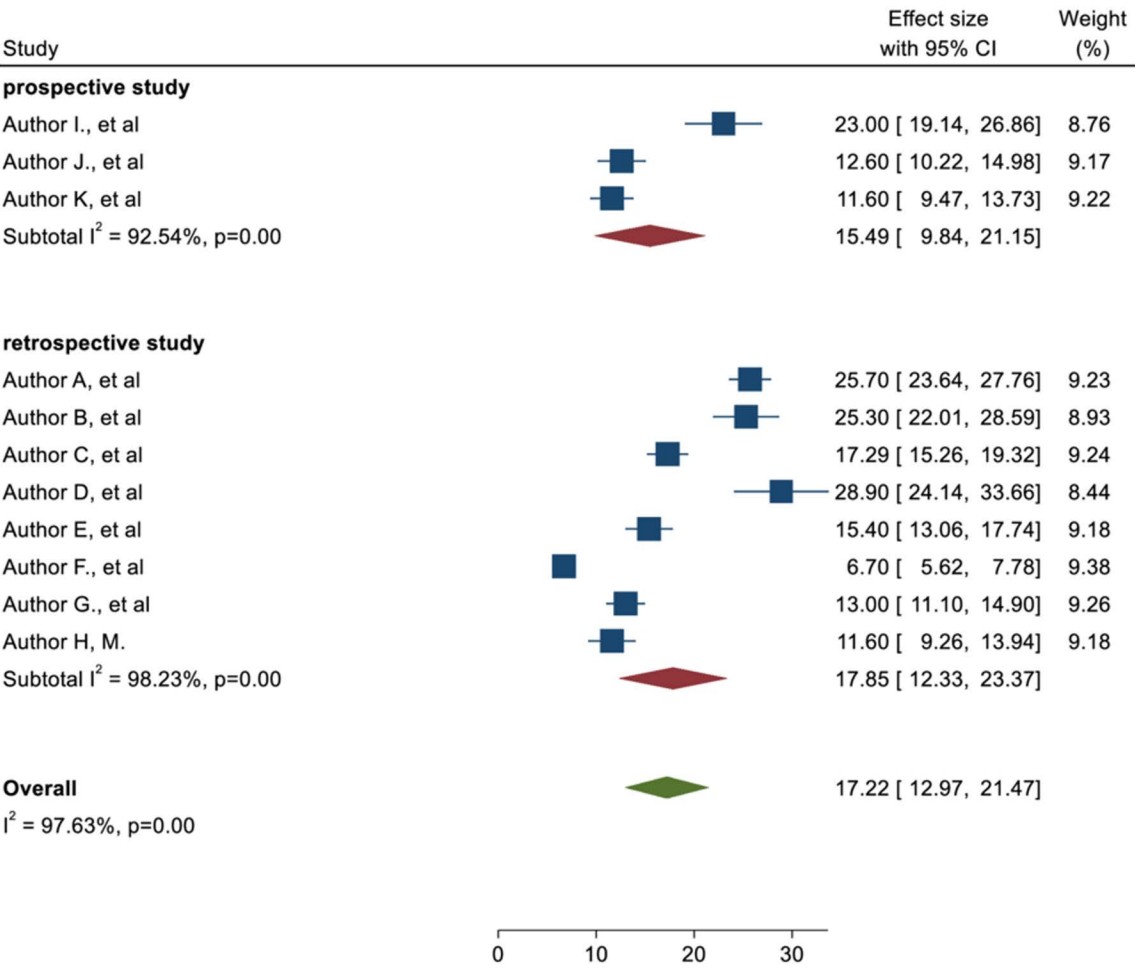

**A) ICU obstetric admission based on study design**

**Fig 4. Subgroup analysis of obstetric admission and maternal mortality in ICU in Africa.**

### 3.8. Indications of obstetrics admission in the ICU in Africa

In this systematic review and meta-analysis, hypertensive disorders of pregnancy emerged as the most common reason for ICU admission among obstetric patients in Africa, accounting for approximately 42.96% (95% CI: 27.36, 58.56; $I^2$=98.35) of patients. This finding underscores the significant burden of hypertensive disorders on maternal health. Obstetric hemorrhage, including both antepartum and postpartum hemorrhage, was identified as the second most common cause of ICU admission, affecting approximately 24.15% (95% CI: 18.12, 30.18; $I^2$=90.84) of patients. (Table 3).

## 4. Discussion

According to this systematic review and meta-analysis, the pooled proportion of obstetric ICU admissions in Africa was 17.22% (95% CI: 12.97–21.47; $I^2$ =97.63%). Even though there was no analogous systematic review and meta-analysis study conducted on this specific research question in the study area, the pooled proportion of obstetric admission in the ICU in this

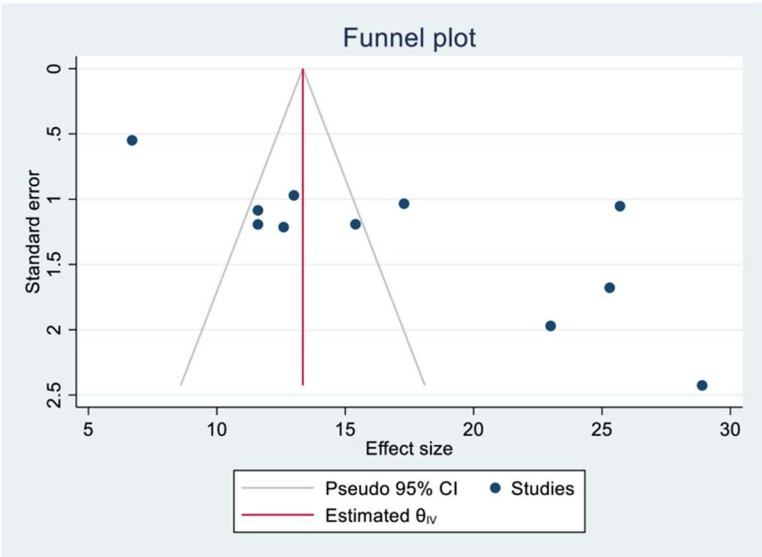

**Fig 5. Funnel plot to test publication bias of 11 studies.**

**Table 2. Sensitivity analysis of the pooled proportion of obstetric admission in ICU in Africa.**

| Omitted study | Proportion | [95% CI] | P-value |
|---|---|---|---|
| Author A, et al. [20] | 16.29 | 12.48,20.11 | 0.000 |
| Author B, et al. [21] | 16.42 | 12.14, 20.69 | 0.000 |
| Author C, et al. [22] | 17.23 | 12.57, 21.89 | 0.000 |
| Author D, et al. [11] | 16.14 | 11.87, 20.41 | 0.000 |
| Author E, et al. [23] | 17.42 | 12.76, 22.08 | 0.000 |
| Author F., et al.[8] | 18.25 | 14.57, 21.94 | 0.000 |
| Author G., et al. [24] | 17.67 | 12.88, 22.46 | 0.000 |
| Author H. [25] | 17.80 | 13.14, 22.47 | 0.000 |
| Author I., et al. [9] | 16.66 | 12.27, 21.06 | 0.000 |
| Author J., et al.[26] | 17.70 | 13.03, 22.37 | 0.000 |
| Author K, et al. [27] | 17.80 | 13.10,22.52 | 0.000 |

**Table 3. Indications for obstetric admission in the ICU in Africa.**

| Indications | Pooled proportion with (95% CI) | $I^2$ | P-value |
|---|---|---|---|
| Hypertensive disorders of pregnancy | 42.96 (27.36–58.56) | 98.35 | 0.000 |
| Obstetric hemorrhage | 24.15 (18.12–30.12) | 90.84 | 0.000 |
| Sepsis | 12.28 (7.54–17.03) | 96.42 | 0.0000 |
| Other non-obstetric causes | 15.38 (9.18–21.59) | 95.16 | 0.0000 |

Other non-obstetrics causes (anesthesia-related complications, pulmonary edema, fatty liver of pregnancy, anemia, coagulopathy, and kidney injury)

study is comparable with the systematic review report done in developed and developing countries, which ranges from 0.4–16.0% since the upper limit (16.0%) indicates ICU obstetric admission in developing countries [28].

The finding of this study was higher than those of studies done in India: 0.41% [29], 9.4% [30], and 2.09% [31], and in eastern Saudi Arabia: 1% [32]. The possible justification for this could be attributed to differences in defining major morbidity criteria for ICU admission and the availability of an alternative facility for intermediate care [29]. Most of the health settings in the included studies of our systematic review may lack a High Dependency Unit (HDU) where women are assessed based on the requirement of basic support, meaning that all cases might transfer to the ICU but the introduction of HDU reduces adult ICU admission by 5% [30]. Furthermore, a high percentage may be related to socio-economic factors such as poverty, malnutrition, low levels of education, poor general health status, delays in seeking health care, and inaccessibility of health services. Cultural and social factors, such as beliefs and practices surrounding childbirth, can also influence the timing and quality of care-seeking behaviors [33].

The overall pooled maternal mortality among obstetric ICU admissions in Africa was 30.69% (95% CI: 23.16, 38.22; $I^2$=93.34%). This high mortality rate could be due to late referral from the peripheral centers, lack of awareness about the disease severity by the community, delay in transportation, and delay in initiation of the treatment [29]. The finding is higher than the review done in the United States and peer nations, which ranged from 0.2% to 9.7% [34]. The reason for this disparity could be the study done in the United States and peer nations focused on countries with higher performance scores which had a certain level of healthcare infrastructure and capacity to provide obstetric care [34]. It is also higher than other studies done in India: 16.6 [31], eastern Saudi Arabia: 9.4% [32], Hong Kong: 3% [35], and Netherland: 4.9% [36]. This could be due to the difference in socio-economic status that is developed nations' lower rates of maternal mortality may be in part due to their well-equipped healthcare settings and ICUs, and high-quality obstetric care services [37]. However, the finding is lower than the studies done in India: 33.8% [29] and 33.3% [30], and Sub-Saharan Africa: 48% [38]. The difference could be due to the difference in the study period that is our review included studies conducted from 2015 onward in this time there is improvement of the maternal health care quality and advancement of health care accesses.

The subgroup analysis based on publication year showed that a higher proportion of obstetric admission and maternal mortality in ICU in studies published after 2020 as compared to those published before 2020. This trend might be attributed to several factors, including the impact of the COVID-19 pandemic, improved surveillance and reporting systems, and advancements in critical care medicine. The pandemic placed significant strain on healthcare systems worldwide, leading to resource shortages and disruptions in care delivery. The lockdowns and restrictions on movement may have limited access to prenatal care and emergency obstetric services, especially in low-resource settings. This could have resulted in delayed or suboptimal care for pregnant women, increasing the risk of complications, ICU admissions, and mortality [39,40]. Additionally, improved awareness and reporting of maternal morbidity and mortality may have contributed to the observed increase in ICU admissions [1]. Furthermore, while improved access to surgical interventions, such as cesarean section, has enhanced maternal outcomes, the overuse or misuse of it can contribute to maternal morbidities (hemorrhage, infection, and thromboembolic events) and potentially increase the risk of complications requiring ICU-level care. Finally, the increased availability of essential obstetric care interventions, such as blood transfusions, magnesium sulfate for pre-eclampsia, and critical care medications, while undoubtedly improving maternal outcomes, may also contribute to a higher proportion of women with severe complications requiring ICU admission [41].

Subgroup analysis based on study design revealed that retrospective studies tended to report a higher proportion of ICU admissions, while prospective studies showed a higher proportion of maternal mortality. This discrepancy may be explained by the nature of the study designs. Retrospective studies, relying on past medical records, might be susceptible to recall bias, leading to underreporting of less severe cases. This could skew the results towards more severe cases requiring ICU admission. Additionally, retrospective studies may have selection bias, as they may be more likely to include cases with significant outcomes [42,43]. In contrast, prospective studies often involve more rigorous data collection methods and regular follow-ups [44], leading to a more accurate identification and reporting of maternal deaths. These studies may also be designed to specifically investigate severe maternal outcomes, potentially contributing to the higher proportion of reported maternal mortality. Furthermore, studies with significant findings, such as higher ICU admissions and maternal mortality rates, may be more likely to be published in recent years [45].

The subgroup analysis revealed regional disparities in the prevalence of obstetric ICU admissions and maternal mortality across Africa. While West Africa had a higher proportion of ICU obstetric admissions as compared to East Africa, East Africa had higher rates of maternal mortality as compared to South Africa. These regional differences may be attributed to several factors: socioeconomic status, healthcare infrastructure and resource allocation, cultural beliefs, and practices [46]. Furthermore, variations in diagnostic criteria and case definitions for severe maternal morbidity may have influenced the identification and reporting of cases. HDP was the predominant indication for ICU admission (43%), which is higher than the study done in India: 25.9% [31] and 38.5% [47], Nepal: 25% [48], Bangalore: 29% [49], and Poland: 37% [50]. The possible justification could be regional variation in the prevalence of hypertensive disorders, which can be influenced by factors such as dietary habits, socioeconomic factors, and genetic predisposition. In certain regions, hypertensive disorders may be more prevalent and severe, leading to a higher proportion of ICU admissions. Furthermore, the availability and accessibility of quality prenatal care, including regular blood pressure monitoring and effective management of gestational hypertension, can vary considerably across different settings. Variations in the characteristics of the study populations, such as age, pre-existing medical conditions, and risk factors for HDP, may also contribute to these differences. Finally, variations in diagnostic criteria and case definitions for HDP across studies may influence the observed rates [51].

Obstetric hemorrhage (24%) was the second most common indication for ICU admission. This is lower than the study done in India: 29.6% [31], Bangalore: 44% [49], and Nepal: 31% [48], but higher than the study in Iran: 19% [37]. This could be due to methodological differences between studies, such as variations in inclusion criteria, data collection methods, and definitions of obstetric ICU admission, which may contribute to the observed disparities [37]. Additionally, access to and quality of emergency obstetric care, including access to blood transfusion services, skilled healthcare providers, and availability of essential medications, can significantly impact the management and outcomes of obstetric hemorrhage. Non-obstetric causes also contribute to 15% of ICU admissions, which include anesthesia-related complications, pulmonary edema, fatty liver of pregnancy, anemia, coagulopathy, and kidney injury. Sepsis was another common indication for 12% of ICU obstetric admissions, which is comparable with the study conducted in India, which was 11% [31]. But higher than the study done in Bangalore, which was 8% [49]. This suggests that sepsis and non-obstetric causes remain a frequent cause of maternal morbidity and mortality, requiring intensive care. The prevalence of risk factors for sepsis, such as underlying medical conditions, malnutrition, and poor hygiene practices, may cause this burden.

This study has several limitations that should be considered when interpreting the findings. First, its reliance on hospital-based data from a limited number of African countries poses a significant challenge to generalizability. The diverse healthcare systems and socioeconomic conditions across the continent mean that these data may not be representative of the true burden of obstetric complications and ICU admissions. Women who do not seek or have access to hospital care, particularly in rural areas or regions with limited healthcare infrastructure, are excluded, potentially leading to a selection bias and an underestimation of the actual rates. This hospital-centric approach may also skew the data towards more severe cases, as these are more likely to require hospital admission.

Second, while the review initially aimed to include studies from 2009 to 2023, the final dataset comprised studies exclusively from 2015 onwards. This truncation of the search period may have excluded relevant data from earlier years, potentially missing important trends or changes in obstetric ICU admissions over time. This limitation also restricts the ability to draw conclusions about the evolution of these trends. Third, the absence of existing systematic reviews on this topic necessitated reliance on individual primary studies for comparison. While this approach was necessary, it introduces heterogeneity as primary studies can vary significantly in methodology, sample populations, data collection tools, and contextual factors. This variability makes direct comparisons challenging and may limit the robustness of the conclusions drawn. A meta-analysis, while performed, is still limited by the quality and consistency of the included primary studies.

Fourth, the exclusion of studies published in languages other than English and those employing different metrics for the primary outcome (specifically, the proportion of obstetric patients among total deliveries) further constrains the scope of this review. This language and measurement bias may have excluded valuable data, particularly from Francophone or Lusophone African countries, and limited the comprehensiveness of the analysis. This could mean overlooking important regional variations or alternative perspectives on the admission landscape. Fifth, the predominance of retrospective studies among the included literature presents another limitation. Retrospective studies, which rely on existing hospital or ICU databases and medical records, are inherently susceptible to limitations in data accuracy and completeness. Challenges in accurately identifying cases, variations in record-keeping practices, and potential loss of information can lead to misclassification and underreporting of obstetric ICU admissions. This limitation could affect the precision of the estimated proportions and potentially underestimate the true burden. Finally, the handling of missing data through substitution methods, while a common practice, introduces a degree of uncertainty into the analysis. The choice of substitution method and the extent of missing data can influence the results and should be considered when interpreting the findings.

In summary, while this study provides valuable insights into obstetric ICU admissions in Africa, these limitations, particularly the reliance on hospital-based data, the limited geographic representation, the language and measurement bias, the retrospective nature of the studies, and the handling of missing data, suggest that caution should be exercised when generalizing the results to the broader African context. Future research addressing these limitations, such as community-based studies, prospective data collection, and inclusion of studies in multiple languages, is needed to provide a more comprehensive understanding of this critical issue.

## 5. Conclusion

The pooled data on obstetric admissions and maternal mortality rates in African ICUs reveals a pressing concern. Hypertensive disorders of pregnancy and obstetric hemorrhage emerge as the predominant causes of these admissions, followed by non-obstetric causes

and sepsis. The substantial proportion of obstetric ICU cases and the associated adverse maternal outcomes underscore the urgent need for dedicated, well-equipped obstetric ICUs across the continent. Investment in the establishment of these facilities, coupled with enhanced emergency obstetric care services, is crucial, particularly in resource-constrained settings. Such investment should prioritize regions with lower socioeconomic status, where maternal mortality rates remain disproportionately high. It is also imperative to implement early warning systems that can effectively identify women at risk for severe obstetric complications. Early recognition and management of obstetric complications, will be vital in improving patient outcomes. Monitoring key performance indicators, including maternal mortality rates will provide valuable insights into the effectiveness of ICU admissions.

Beyond infrastructure and systems strengthening, proactive measures are essential. This includes widespread implementation of evidence-based interventions such as preeclampsia screening and prevention, along with launching community awareness campaigns to educate expectant mothers and their families about the signs and symptoms of obstetric complications, which are critical. Empowering women to seek timely care and ultimately lead to improved outcomes. These initiatives align with the WHO's Maternal Health Strategy, and Sustainable Development Goals can contribute directly to national efforts to reduce maternal mortality. For example, scaling up preeclampsia screening programs could be integrated into existing antenatal care services and can leverage existing healthcare infrastructure and personnel. Likewise, community awareness campaigns could be implemented through partnerships with local organizations and community health workers, ensuring culturally appropriate and accessible messaging. These interventions should be prioritized within national health strategies and adequately funded to ensure widespread coverage and impact.

The study's reliance on hospital-based data and the heterogeneity of healthcare systems across the included countries restricts the generalizability of our findings and highlights the need for future large-scale, population-based studies to accurately assess the burden of disease and explore regional variations. Such research, along with prospective studies evaluating specific interventions, is crucial to develop targeted and effective strategies to improve maternal health in Africa. Specifically, future research should investigate the effectiveness of different models of obstetric care delivery, including community-based interventions and tele-health strategies, to inform policy decisions on resource allocation and service delivery. Given the limitations of the available data, policymakers should prioritize investments in strengthening data collection systems, improving access to facility-based care for all women, particularly in underserved communities, and supporting the implementation and evaluation of evidence-based interventions. This multifaceted approach is essential for fostering a safer environment for maternal health in Africa.

## Implications of the study

The high proportion of obstetric ICU admissions in Africa highlights a significant public health problem.

The findings highlight the need for improved access to quality maternal healthcare, particularly in resource-limited settings.

The study emphasizes the importance of early identification and management of obstetric complications to prevent severe outcomes and reduce the need for ICU admission.

## Supporting information

**S1 File. Critical appraisal checklist.**
(DOCX)

**S2 File. PRISMA checklist.**
(DOCX)

**S3 File. Search strategy for the selected databases.**
(DOCX)

**S4 File. Extraction tool.**
(XLSX)

**S5 File. List of included and excluded studies.**
(DOCX)

## Author contributions

**Conceptualization:** Alemwork Abie, Molla Getie Mehari, Tenagnework Eseyneh Dagnew, Amare Mebrat Delie, Mihret Melese.

**Data curation:** Mickiale Hailu.

**Formal analysis:** Alemwork Abie, Liknaw Workie Limenh.

**Funding acquisition:** Liknaw Workie Limenh.

**Methodology:** Alemwork Abie, Liknaw Workie Limenh.

**Software:** Mickiale Hailu.

**Supervision:** Mickiale Hailu.

**Validation:** Molla Getie Mehari, Tenagnework Eseyneh Dagnew, Amare Mebrat Delie, Mihret Melese.

**Visualization:** Tenagnework Eseyneh Dagnew, Amare Mebrat Delie, Mihret Melese.

**Writing – original draft:** Alemwork Abie, Nigus Kassie Worku, Eneyew Talie Fenta, Dereje Esubalew.

**Writing – review & editing:** Molla Getie Mehari, Tenagnework Eseyneh Dagnew, Amare Mebrat Delie, Mihret Melese, Nigus Kassie Worku, Eneyew Talie Fenta, Dereje Esubalew, Mickiale Hailu.

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
