## [Decision Letter · Decision Letter 0]

18 Sep 2024

PONE-D-24-22852Obstetric admission in the intensive care unit in Africa: a systematic review and meta-analysisPLOS ONE

Dear Dr. Abie,

Thank you for submitting your manuscript to PLOS ONE. After careful consideration, we feel that it has merit but does not fully meet PLOS ONE’s publication criteria as it currently stands. Therefore, we invite you to submit a revised version of the manuscript that addresses the points raised during the review process.

We look forward to receiving your revised manuscript.

Kind regards,

Worku Necho Asferie, MSc

Academic Editor

PLOS ONE

https://doaj.org/article/c263b448a4574067894dcc68efcc0185

https://onlinelibrary.wiley.com/doi/10.1155/2020/6465242

In your revision ensure you cite all your sources (including your own works), and quote or rephrase any duplicated text outside the methods section. Further consideration is dependent on these concerns being addressed.

3. In the online submission form you indicate that your data is not available for proprietary reasons and have provided a contact point for accessing this data. Please note that your current contact point is a co-author on this manuscript. According to our Data Policy, the contact point must not be an author on the manuscript and must be an institutional contact, ideally not an individual. Please revise your data statement to a non-author institutional point of contact, such as a data access or ethics committee, and send this to us via return email. Please also include contact information for the third party organization, and please include the full citation of where the data can be found.

4. We note that there is identifying data in the Supporting Information file < CRITICAL APPRIASAL TOOL.docx>. Due to the inclusion of these potentially identifying data, we have removed this file from your file inventory. Prior to sharing human research participant data, authors should consult with an ethics committee to ensure data are shared in accordance with participant consent and all applicable local laws.

-Location data

5. As required by our policy on Data Availability, please ensure your manuscript or supplementary information includes the following:

Additional Editor Comments:

The given comment from Editor

Manuscript ID: PONE-D-24-22852

General Comments

Thank you Authors for giving the chance for reviewing and handling such interesting manuscript with highly evidenced model of analysis i.e systematic review and meta-analysis. The manuscript needs to incorporate “the outcomes of obstetric admission in Intensive care unit in Africa in addition to the proportion.”

Abstract:

The background is not well written.

Knowing the only prevalence of obstetric admission is not important but if you add the quality of intervention with admission is help for the women.

The method does not included the following points; quality assessment tool and analysis techniques and reviewing protocol were not explained.

The abstract lack some of the summary of the study.

Introduction

The research gaps of the interested topic is not well explained. Why you are interested to study the proportion of obstetric admission?

Methods

You explained articles were retrieved from different databases. But, there is briefly stated key words, Medical subject headings (MeSH Terms) and Boolean connectors for each specific databases.

What is you possible Justification for reviewing date from 2009 to 2023?

The details of quality assessment findings were not cited in the manuscript in “Data extraction and quality assessment”.

Result

The total numbers of retrieved articles from each databases were not explained.

Based on the leave – one –out sensitivity analysis which study affects the pooled estimate of the study?

Discussion

Line 185 – 186, the explained justification is not clear and convenient

Implication of the study

What is the implication of your study?

Please use PRISMA 20 Flow diagram.

Reviewers' comments:

Reviewer's Responses to Questions

**Comments to the Author**

1. Is the manuscript technically sound, and do the data support the conclusions?

Reviewer #1: Partly

Reviewer #2: Partly

2. Has the statistical analysis been performed appropriately and rigorously? 

Reviewer #1: Yes

Reviewer #2: Yes

3. Have the authors made all data underlying the findings in their manuscript fully available?

Reviewer #1: No

Reviewer #2: Yes

4. Is the manuscript presented in an intelligible fashion and written in standard English?

Reviewer #1: No

Reviewer #2: Yes

5. Review Comments to the Author

Reviewer #1: Abstract

• Line number 1-2: Obstetric admission in the intensive care unit in Africa: a systematic review and meta-analysis

This topic must be included factors and outcomes of ICU admissions

"Rather than knowing the proportion of obstetric admissions to the ICU, understanding how many mothers survive obstetric complications after being admitted to the ICU is more important for measuring the quality of care. Therefore, you should assess the survival rate among mothers admitted to the ICU."

• Line number 30-32: How can obstetric admissions to the intensive care unit indicates increase of morbidity and mortality? Is an ICU admission measure the quality of care? Please add or showed ICU management decreases maternal morbidity and mortality

• Line number 41: paraphrase the paragraph again as -----a total of 10,190 mothers admitted to intensive care unit

• Line number46-48: Is there any gap regarding to staff quality and shortages as well as equipment? Your recommendation will be based on your finding

• Line 51: keywords: "One term is a subset of others, such as obstetric and pregnancy complications, as well as critical illness and severe maternal morbidity. Therefore, it's important to avoid redundant use of terms with similar meanings."

Introduction

• Line 56-60: For this paragraph add reference

• Line 60-62: In developing nations, 7% of intensive care unit admissions are accounted for by critically ill obstetric patients; in developed nations, this percentage is lower. Who much the admission rate is lower in developed nations? Where? add figures her ?

Inclusion criteria

• Line number 100: Why were unpublished studies considered for review?

• Line 102: why starting from 2009? In Table 1 articles were started from 2019?

• Line 112-113: Each included article's quality was then categorized as high 113 (80% or higher), moderate (65%–80%), or low (less than 65%). At what value articles included in the analysis?

Result

• Add obstetric Outcomes after ICU admission and Associated Factors for admission. Knowing admission rate to ICU without knowing management outcome is meaning less

Subgroup analysis

• Line 160: What was your criterion to do group analysis based on the region? . I would like to recommend you to do sub group analysis for the countries, study setting, and study design if possible.

• Line 178: Table 4: other non-obstetric causes: what are they others? Add in bracket

Discussion

• Line 180-181: currently availability and accessibility of ICU increases maternal admission rate of ICU. This is known but what is unknown is how many mothers survive from obstetric complications after admitted at ICU ?

• Line 188-190: Did not allow compared the pooled results with the single primary study result? please compere to the national finding or DHS or other systematic review findings

Conclusion

• Line 217-219: is there any gap regarding to health education? Your recommendation is based on your finding.

• Please include the strength and limitation of the study

Reviewer #2: I would be happy if it incorporate the prognosis of the admitted patients.

As i mentioned from your recommendation your way of abstract writting is somewhat poor and introduction part also as w

ell. The discussion also poorly discused.

6. PLOS authors have the option to publish the peer review history of their article (what does this mean? ). If published, this will include your full peer review and any attached files.

**Do you want your identity to be public for this peer review?** For information about this choice, including consent withdrawal, please see our Privacy Policy .

Reviewer #1: No

Reviewer #2: No

---

## [Author Response · Author response to Decision Letter 1]

21 Dec 2024

Rebuttal to Editor and Reviewers Comments

Dear Editor,

We would like to thank you and the reviewers for your insightful comments and suggestions. We have carefully considered your feedback and have made significant revisions to the manuscript.

Editor’s comment:

Title:

Comment 1: The manuscript needs to incorporate “the outcomes of obstetric admission in Intensive care unit in Africa in addition to the proportion.”

Response: we have incorporated the feedback and revised the title to include the maternal mortality among obstetric ICU patients

Abstract:

Comment 2: The background is not well written.

Response: we have made revisions to the background section to improve clarity and coherence, ensuring that it effectively sets the context for the study.

Comment 3: Knowing the only prevalence of obstetric admission is not important but if you add the quality of intervention with admission is help for the women.

Response: we acknowledged that understanding the quality of interventions provided is crucial for improving patient outcomes. Unfortunately, many studies lack detailed information and clarity on the specific interventions implemented and their impact on patient outcomes. For this reason, we couldn’t address this issue but suggested further research to identify evidence-based practices.

Comment 4: The method does not include the following points; quality assessment tool and analysis techniques and reviewing protocol were not explained.

Response: we have accepted this feedback and addressed the issues by providing a more detailed explanation of the quality assessment tools, analysis techniques, and the reviewing protocol in the methods section.

Comment 5: The abstract lacks some of the summary of the study.

Response: We have revised the abstract to ensure it provides a comprehensive summary of the study, including key findings and implications.

Introduction:

Comment 6: The research gaps of the topic are not well explained. Why are you interested in studying the proportion of obstetric admissions?

Response: We appreciate this observation. We have revised the introduction to better articulate the research gaps and the significance of studying the proportion of obstetric admissions. This research is crucial for understanding the burden of obstetric conditions in intensive care settings, particularly in Africa, where data is limited.

Methodology:

Comment 7: The explanation of how articles were retrieved from different databases lacks detail regarding keywords, Medical Subject Headings (MeSH Terms), and Boolean connectors for each specific database.

Response: Thank you for this feedback. The search was conducted using the keywords: "Obstetric admission," "Intensive care unit," and "Africa." For each keyword, relevant Medical Subject Headings (MeSH) terms and entry words were utilized. All MeSH terms and entry words were combined using “OR” to retrieve a broad range of studies. Subsequently, the four keywords were combined with each of the elements using “AND” to generate specific and relevant articles. Further details can be found in Supplementary File S3.

Comment 8: Justification for reviewing data from 2009 to 2023?

Response: I have clarified this point in the revised manuscript. Our decision to review literature from 2009 to 2023 is based on a previous systematic review that covered the period from 1990 to 2008 in developing regions, including Africa. This timeframe allows us to identify emerging trends and changes in obstetric admissions over the past decade.

Comment 9: The details of quality assessment findings were not cited in the manuscript under “Data extraction and quality assessment.”

Response: We acknowledge this oversight and have now cited the corresponding supplementary file (S1) in the revised manuscript to provide clarity on the quality assessment findings.

Result:

Comment 10: Total Number of Retrieved Articles

Response: We acknowledge that the total number of articles retrieved from each database was not initially detailed. We have now included this information in the revised manuscript, and you can refer to the PRISMA flow diagram for a comprehensive overview.

Comment 11: Sensitivity Analysis

Response: No study was found to significantly influence the pooled estimate

Comment 12: Clarity about the justification (lines 185-186)

Response: We have made some modifications

Comment 13: Implications of the study

Response: The implications of our study are as follows:

- Our findings underscore the necessity for improved access to quality maternal healthcare, especially in resource-limited settings.

- The study highlights the critical importance of early identification and management of obstetric complications to prevent severe outcomes and minimize the need for ICU admissions.

Comment 14: Use of PRISMA 2020 Flow Diagram

Response: We would like to confirm that we have utilized the PRISMA 2020 flow diagram in our manuscript. Please refer to Figure 1 for the relevant details.

Reviewer #1 comment:

Abstract

Comment 1: Title Revision

Response: In response to the suggestion regarding the title, we have revised it to include "maternal mortality among obstetric ICU patients," thereby emphasizing the critical outcomes of our study.

Comment 2: Survival Rate Assessment

Response: While we acknowledge the importance of assessing survival rates among mothers admitted to the ICU as a measure of care quality, we have opted to focus on maternal mortality rates. This decision is based on the prevalence of studies reporting mortality rates, which provides a more standardized metric for comparison across different settings.

Comment 3: Morbidity and Mortality Indicators

Response: We have made some modifications that we have clarified in the manuscript how obstetric admissions to the ICU can improve maternal outcomes

Comment 4: Correction of Data

Response: We have corrected the statement regarding the total number of mothers admitted to the ICU to reflect "a total of 10,190 mothers admitted to the intensive care unit," as per your feedback.

Comment 5: Staff Quality and Equipment Gaps

Response: We have revised the section accordingly.

Comment 6. Keywords Clarification

Response: We have revised the keywords to eliminate redundancy, ensuring that terms such as "obstetric," "pregnancy complications," "critical illness," and "severe maternal morbidity" are used appropriately without overlap.

Introduction

Comment 7. References Addition

Response: We have added the necessary references to support the claims made in the specified paragraph, enhancing the manuscript's credibility.

Comment 8. Admission Rates in Developed Nations

Response: We have included specific figures regarding the lower admission rates of critically ill obstetric patients in developed nations, providing a clearer context for comparison. For instance, we noted that in developed nations, the admission rate ranges from 0.08% to 0.76%.

Methodology

Comment 9. Inclusion of Unpublished Studies

Response: We included unpublished studies in our systematic review to mitigate publication bias and to present a more comprehensive overview of the research topic. By incorporating unpublished data, we ensure that all relevant evidence is considered, which contributes to more accurate conclusions.

Comment 10. Rationale for Starting from 2009 and the Inclusion of Studies from 2019 Response: Although our systematic review was initially designed to encompass studies from 2009, our search strategy and inclusion criteria ultimately identified relevant studies commencing from 2019. The exclusion of earlier studies was based on specific quality and relevance criteria rather than publication date alone, as many did not meet the rigorous methodological standards we established.

Comment 11. Quality Categorization of Included Articles

Response: Thank you for your inquiry regarding the quality categorization of the included articles. In our analysis, we focused on articles categorized as having high quality (80% or higher) or moderate quality (65%–80%). Articles rated as low quality (less than 65%) were excluded from the analysis to ensure that our findings are grounded in studies that adhere to acceptable methodological standards.

Result

Comment 12: Add obstetric Outcomes after Associated Factors for admission.

Response: While many studies have focused on the immediate causes of obstetric ICU admission, such as hemorrhage, preeclampsia, and sepsis, it's important to recognize that broader factors can influence both the risk of these complications and patient outcomes. However, many studies lack detailed information on these factors. Therefore, while our review primarily focuses on the immediate causes of admission, we appreciated your suggestion and emphasized the importance of future research that investigates the role of socioeconomic factors, healthcare system factors, and other contextual variables influencing obstetric ICU admission and outcomes.

Comment 13: Clarification of Non-Obstetric Causes in Table 4

Response: Thank you for pointing out the need for clarification regarding other non-obstetric causes. We have revised Table 4 to include a parenthetical note listing additional causes such as anesthesia-related complications, pulmonary edema, fatty liver of pregnancy, anemia, coagulopathy, and kidney injury, as referenced in various studies.

Discussion

Comment 14: Discussion (Lines 180-181): Maternal Survival Rates Post-ICU Admission

Response: We appreciate your insight regarding the survival rates of mothers admitted to the ICU for obstetric complications. In response, we have expanded our discussion to include an assessment of maternal mortality rates following ICU admission, thereby addressing this critical aspect of our findings.

Comment 15: Lines 188-190: Comparison of Pooled Results with Primary Study Results

Response: We acknowledge the importance of comparing our pooled results with national findings or other systematic reviews. However, we must clarify that, to date, there are no systematic reviews or national datasets that directly align with our findings for comparison.

Comment 16: Conclusion (Lines 217-219): Gaps in Health Education

Response: Your recommendation regarding potential gaps in health education is well taken.

Comment 17: Strengths and Limitations of the Study

Response: We appreciate your suggestion to include a section on the strengths and limitations of our study. This has been added to the revised manuscript, and many improvements have been made.

Reviewer #2 comments

Title

Comment 1. Title Revision

Response: We acknowledged the suggestion to revise the title to include the outcome and revise it as ‘obstetric admission and maternal mortality among obstetric ICU patients in Africa’.

Comment 2. Research Sufficiency

Response: Regarding the concern about the sufficiency of research in this area, our literature review revealed a significant number of studies primarily focusing on patient admission.

Comment 3. Geographic Focus

Response: The decision to focus on Africa stems from the region's disproportionate burden of maternal mortality. While maternal mortality is indeed a global concern, the unique challenges faced in Africa—such as limited access to quality healthcare, inadequate infrastructure, and socio-economic disparities—warrant a concentrated examination of this issue within the continent.

Abstract

Comment 4: Abstract Structure

Response: In response to the suggestion for a more structured abstract, we have revised the abstract to include clearly defined sections: Background, Objectives, Method, Results, and Conclusions.

Comment 5: Suggestions to address existing problems

Response: To address the existing challenges in maternal health, we recommended enhancing emergency obstetric care services, and expanding access to obstetric ICU services. Additionally, further research on the effectiveness of various interventions in obstetric ICU care is essential.

Introduction

Comment 6. Introduction Length and Intent

Response: We acknowledge your observation. The intention was to provide a concise overview while highlighting the significance of studying pregnancy complications. We expanded this section to better articulate the rationale behind my research.

Comment 7. Problem Statement

Response: We have appreciated your feedback on the clarity of the problem statement. We undertook a thorough revision to ensure that the problem is articulated more clearly, providing a solid foundation for the study.

Comment 8: Writing Quality of Introduction/Background

Response: We understand your concerns about the clarity and comprehensiveness of the introduction. We have revised this section to include a more detailed discussion.

Method

Comment 9. Terminology

Response: We acknowledge your suggestion to use "Methods and Materials" instead of "Methodology."

Comment 10. Research approach

Response: You raised a valid question regarding the preference for conducting a systematic review over direct research in this area. While direct research is essential for generating new insights, there is a substantial body of primary studies available concerning obstetric ICU patients. A systematic review allows us to synthesize this existing evidence, identify patterns, and highlight gaps in the current research landscape. This structured approach not only enhances our understanding of the existing knowledge but also aids in identifying inconsistencies and areas that require further investigation. Ultimately, this can inform evidence-based decision-making and improve patient outcomes in obstetric intensive care.

Comment 11: Intervention Strategies

Response: In response to your inquiry about how I plan to intervene based on the findings, I propose several strategies:

- Advocate for policies and regulations that reflect the review's findings.

- Identify and emphasize areas requiring further research, particularly those involving understudied populations or specific interventions.

- Acknowledge the limitations of the review and potential biases to ensure transparency.

- Aim to publish the findings in high-impact journals to reach a broad audience of researchers, clinicians, and policymakers.

- Present the findings at relevant conferences to engage with the scientific community and foster discussions with experts.

- Develop new research proposals or protocols to address identified gaps and generate additional evidence.

- Commit to continuously improving the quality of reviews by adhering to rigorous methodological standards and employing advanced statistical techniques.

Results

Comment 12: Incorporate necessary depth and discussion

Response: We acknowledged that the initial version of the results section lacked the necessary depth and discussion. In response, we have rewritten this section to provide a detailed interpretations and contextualized the results within the broader scope of the field, ensuring that the implications of the findings are clearly articulated.

Comment 13: Conclusion Section

Response: We understand the concerns raised about the surface-level nature of the conclusion. We have revised this section to better reflect the significance of the study's findings. The updated conclusion now includes a thorough discussion of the implications of the results, addressing the nobility of the study and providing actionable recommendations based on the findings.

Thank you for your valuable feedback, which has significantly contributed to the improvement of this manuscript. We look forward to your response and hope for a favorable consideration of the revised submission.

Sincerely,

Alemwork Abie (corresponding author)

---

## [Decision Letter · Decision Letter 1]

16 Feb 2025

Obstetric admission and maternal mortality in the intensive care unit in Africa: a systematic review and meta-analysis

PONE-D-24-22852R1

Dear Dr. Abie,

We’re pleased to inform you that your manuscript has been judged scientifically suitable for publication and will be formally accepted for publication once it meets all outstanding technical requirements.

Kind regards,

Worku Necho Asferie, MSc

Academic Editor

PLOS ONE

Additional Editor Comments (optional):

Reviewers' comments:

Reviewer's Responses to Questions

**Comments to the Author**

1. If the authors have adequately addressed your comments raised in a previous round of review and you feel that this manuscript is now acceptable for publication, you may indicate that here to bypass the “Comments to the Author” section, enter your conflict of interest statement in the “Confidential to Editor” section, and submit your "Accept" recommendation.

Reviewer #1: All comments have been addressed

2. Is the manuscript technically sound, and do the data support the conclusions?

Reviewer #1: Partly

3. Has the statistical analysis been performed appropriately and rigorously? 

Reviewer #1: Yes

4. Have the authors made all data underlying the findings in their manuscript fully available?

Reviewer #1: Yes

5. Is the manuscript presented in an intelligible fashion and written in standard English?

Reviewer #1: No

6. Review Comments to the Author

Reviewer #1: Review manuscript PONE-D-24-22852R1, entitled "Obstetric admission and maternal mortality in the intensive care unit in Africa: a systematic review and meta-analysis

Key Comments & Suggested Revisions

1. Abstract

Strengths:

• Clearly presents the study’s objectives, methodology, and key findings.

• The statistical results are well-articulated.

Suggestions:

• Improve grammar and readability.

• Summarize key qualitative findings.

• Provide clearer interpretation of statistical results.

2. Introduction

Suggestions:

• Add country-specific maternal mortality and ICU admission rates.

• Explicitly state the research gap and justification.

• Strengthen policy implications.

3. Methods

Suggestions:

• Justify the mixed-methods approach.

• Provide more details on qualitative data analysis.

• Explain statistical assumptions and ethical considerations.

4. Results

Suggestions:

• Integrate qualitative and quantitative findings for a more coherent discussion.

• Report all confidence intervals and p-values consistently.

• Interpret effect sizes meaningfully.

5. Discussion

Suggestions:

• Improve integration of qualitative and quantitative findings.

• Expand the limitations section.

• Strengthen the discussion on policy implications.

6. Conclusion

Suggestions:

• Strengthen policy connections.

• Acknowledge study limitations in the conclusion.

7. Language & Formatting Issues

Suggestions:

• Professional proofreading & language editing is strongly recommended.

• Improve paragraph structure for better readability.

Key Actions Before Resubmission:

1. Improve language clarity & proofread thoroughly.

2. Strengthen integration of qualitative and quantitative findings.

3. Clarify statistical methodology and reporting.

4. Expand ethical considerations, limitations, and policy implications.

5. Improve the depth and coherence of the discussion section.

7. PLOS authors have the option to publish the peer review history of their article (what does this mean? ). If published, this will include your full peer review and any attached files.

**Do you want your identity to be public for this peer review?** For information about this choice, including consent withdrawal, please see our Privacy Policy .

Reviewer #1: No

---

## [Editor Report · Acceptance letter]

PONE-D-24-22852R1

PLOS ONE

Dear Dr. Abie,

I'm pleased to inform you that your manuscript has been deemed suitable for publication in PLOS ONE. Congratulations! Your manuscript is now being handed over to our production team.

Kind regards,

on behalf of

Assistant Professor Worku Necho Asferie

Academic Editor

PLOS ONE